# Smartphone Use Is Associated with Low Prevalence of Locomotive Syndrome among Elderly Individuals with Musculoskeletal Disorders

**DOI:** 10.3390/ijerph192316213

**Published:** 2022-12-04

**Authors:** Naoto Miyashita, Tomohiro Ishida, Tatsunori Ikemoto, Atsuhiko Hirasawa, Young-Chang Arai, Masataka Deie

**Affiliations:** 1Department of Orthopedic Surgery, Aichi Medical University, Nagakute 480-1195, Japan; 2Department of Rehabilitation, Aichi Medical University Hospital, Nagakute 480-1195, Japan; 3Multidisciplinary Pain Center, School of Medicine, Aichi Medical University, Nagakute 480-1195, Japan

**Keywords:** smartphone, older adults, locomotive syndrome, musculoskeletal disorders

## Abstract

Objectives: Physical activity management through smartphone applications is increasing worldwide; however, it is unclear whether smartphone users among elderly Japanese individuals with musculoskeletal disorders are less likely to experience “locomotive syndrome” (LoS). We aimed to test the hypothesis that LoS in smartphone users had lower prevalence than that in non-smartphone users among elderly individuals with musculoskeletal disorders. Methods: Elderly participants, aged ≥60 years, who visited the outpatient clinic were enrolled. All participants were asked whether or not they used smartphones and were allocated into either the smartphone group or the non-smartphone group. After completing the 25-question Geriatric Locomotive Function Scale (GLFS-25), LoS prevalence was determined by 3-stage cutoff values of the GLFS-25 score (≥7, ≥16, and ≥24), and the total and three subdomain scores (body pain, movement-related difficulty, and psychosocial complications) were compared between the two groups. Generalized linear regression was then performed to confirm whether the use of smartphones was associated with lower GLFS-25 scores, even after controlling for confounders. Results: Overall, 266 participants, aged ≥60 years, were recruited. LoS prevalence was significantly higher in the non-smartphone group than in the smartphone group at all stages (all *p* < 0.001). Mean GLFS-25 total and subdomain scores were significantly lower in the smartphone group than in the non-smartphone group (all *p* < 0.001), and these statistical relationships were maintained even after controlling for age and sex. Conclusions: Smartphone use was associated with low LoS prevalence and low GLFS-25 scores among elderly individuals with musculoskeletal disorders, although the causal relationship remains unclear.

## 1. Introduction

In an advanced aging society, maintaining a healthy life expectancy for elderly individuals with musculoskeletal disorders remains an important issue, and the Japanese society is facing increasing medical expenses.

There has been considerable development in individual digital equipment, and mobile phones are essential personal equipment in Japanese society. However, there remains a considerable generation gap in the usage of digital devices between younger and older people. Although there are different types of mobile phones (i.e., feature phones or smartphones), smartphones have been recognized as the most popular device because they have many convenient functions, such as counting daily activities and social networking services (SNS) [1]. Smartphone ownership among elderly people varies substantially according to age: 59% of Americans aged 59–65 years were smartphone users, while only 17% of those aged ≥80 years are smartphone users [2]. The main concerns for the use of smartphones in the elderly are financial limitations, visual limitations, lack of interest, and lack of knowledge in using technological devices and advanced functionalities [1]. Although the use of information and communication technology (ICT) is increasing, elderly people still often lack experience with ICT use. However, if ICT can be used effectively, it may help resolve issues related to medical care, nursing care, and health. Physical activity management through smartphone applications (“apps”) is increasing worldwide, and their widespread use is expected to provide new insights for health [3].

The prevalence of musculoskeletal complaints among mobile device users has been reported to range from 1.0% to 67.8%, with neck complaints being the highest at 17.3% to 67.8% [4]. However, there is only limited evidence showing that mobile touchscreen use is associated with musculoskeletal symptoms [5]. As a positive impact of smartphones, using social networking sites to share physical activity has been reported to be positively correlated with social connectedness [6].

The Japanese Orthopedic Association proposed the concept of locomotive syndrome (LoS), which is defined as a decline in locomotor function due to a musculoskeletal disorder, often leading to the need for nursing care [7,8,9]. The concept suggests the importance of allocating the condition of each motor element in the overall mobility function. A screening tool for locomotive syndrome is the 25-question Geriatric Locomotor Functioning Scale (GLFS-25), a self-reported questionnaire and physical assessment scale [10]. These are expected to be used as screening tools to identify those at high risk of losing motor function. With this background, we hypothesized that there might be a possible relationship between smartphone use by elderly people and their locomotive function. To support this, we also hypothesized that the prevalence of LoS in smartphone users was lower than that in non-users among elderly individuals with musculoskeletal disorders. This study aimed to test these hypotheses through a cross-sectional study.

## 2. Materials and Methods

### 2.1. Study Design and Ethics

This was a cross-sectional study approved by the institutional ethics committee of Aichi Medical University [No. 2020-234] and was conducted in accordance with the World Medical Association Declaration of Helsinki. The requirement for written informed consent was waived because the content of the current study was made public through the hospital’s website, and all participants who met the eligibility criteria could check the study details at any time. All anonymous data that did not identify the study participants were allowed to be used only for this study unless the patient refused to provide the information in accordance with the withdrawal strategy.

### 2.2. Enrollment of Participants

Participants were recruited from the outpatient ward of our hospital between April 2019 and December 2020. The evaluators asked patients to answer the GLFS-25 questionnaire to assess their locomotive function. The evaluators also asked patients about the type of mobile phones they were using: smartphone, feature phone, or no mobile phone. Participants were then classified into the smartphone or non-smartphone groups. The latter group included participants who did not use a mobile phone.

Inclusion criteria were as follows: (1) age ≥60 years and (2) complaints of musculoskeletal disorders. Patients were excluded if they (1) were diagnosed with acute traumatic injuries, (2) were unable to walk independently with or without a cane, (3) had confirmed or suspected dementia, and (4) responses to any items in the questionnaire were missing. Questionnaires were collected until a sufficient sample was achieved during the visits.

### 2.3. Assessment of Locomotive Syndrome

A quantitative screening tool called the 25-item GLFS-25 was developed to measure LS severity [11]. GLFS-25 is a self-administered, relatively comprehensive measure that consists of 25 items, including four questions regarding pain during the last month, 16 questions regarding activities of daily living during the last month, 3 questions regarding social functions, and 2 questions regarding mental health status during the last month. These 25 items are graded on a 5-point scale from no impairment (0 points) to severe impairment (4 points) and, subsequently, arithmetically added to produce a total score (minimum 0 and maximum 100). The validity of the GLFS-25 for Japanese people has also been verified in previous studies [11,12,13]. Using the recently determined GLFS-25 cutoff value, participants were divided into three groups. Locomotor function was defined based on the GLFS-25: normal, GLFS-25 score of <7; LoS stage 1 (LoS-1), GLFS-25 score of ≥7 and <16; LoS stage 2 (LoS-2), GLFS-25 score of ≥16 and <24; LoS stage 3 (LoS-3), GLFS-25 score of ≥24 [12]. A higher score indicates worse locomotive function. In addition to the total score, three subdomains of LS (body pain, movement-related difficulty, and psychosocial complications) were investigated based on the method described by Wang et al. [13].

### 2.4. Statistical Analysis

#### 2.4.1. Sample Size

The number of participants was determined by sample size estimation using G*power3 software [14]. In our preliminary sample collection from 40 patients with musculoskeletal disorders, 20 did have a smartphone and 20 did not have a smartphone, and the standardized mean difference of the GLFS-25 score between them was approximately 0.4. Based on this value, at least 133 participants per group were required, with a type I error of 5% and power of 90%.

#### 2.4.2. Differences in Variables between the Two Groups

Continuous variables were presented as mean and standard deviation or median and interquartile range according to the data distribution, whereas categorical variables were presented as number and percentage of patients. Variables between the two groups were compared using the chi-square test, Student’s *t*-test, or Mann–Whitney U test for demographic data, and the prevalence of LoS was determined according to the appropriate fitting models. Generalized linear models were used to identify differences in the GLFS-25 total and subdomain scores between the two groups, adjusting for the potential confounders such as age and sex. Analyses were performed using SPSS software (version 26, SPSS Inc., Chicago, IL, USA). All results were considered to be statistically significant at *p* < 0.05.

## 3. Results

### 3.1. Characteristics of Participants

A total of 266 participants were recruited. The average age of the participants was 74.7 ± 6.4 years (95 males and 171 females). The mean age of patients in the smartphone group was significantly lower than that of the non-smartphone group (*p* < 0.001) (Table 1). Reasons for orthopedic consultation among the participants were osteoarthritis of the hip and knee (39/266; 14.6%), rheumatoid arthritis and polymyalgia rheumatica (82/266; 30.8%), spinal disease (124/266; 46.6%), and others (21/266; 7.8%). There were no significant differences in disease entities between the two groups (Table 1).

### 3.2. Prevalence and Severity of Locomotive Syndrome between the Two Groups

As shown in Table 2, the prevalence of LoS was significantly higher in the non-smartphone group than in the smartphone group at every stage (all *p* < 0.001).

Cronbach’s alpha for the GLFS-25 total score was 0.95, and Cronbach’s alpha for the GLFS-25 subdomain score was 0.82 for body pain, 0.92 for movement difficulty, and 0.94 for psycho-social complication. These results indicate that the internal consistency of this self-reported questionnaire was adequate [15].

Figure 1 shows the GLFS-25 total score for the smartphone and non-smartphone groups. The mean GLFS-25 total score for all participants was 24.9 ± 17.6. The score of the non-smartphone group (30.7 ± 17.5) was higher than that of the smartphone group (19.2 ± 15.7) (*p* < 0.001).

Figure 2, Figure 3 and Figure 4 shows the results of the GLFS-25 subdomain scores for the smartphone and non-smartphone groups divided into three structural factors: body pain, movement-related difficulty, and psychosocial complications. Scores of the smartphone group and the non-smartphone group were 6.4 ± 4.2 and 8.9 ± 4.7 points, respectively, for body pain, 6.4 ± 6.8 and 10.9 ± 7.6 points for movement-related difficulty, and 10.1 ± 9.6 and 17.4 ± 10.5 points for social and daily activity. For all subdomains, the scores for the non-smartphone group were significantly higher than those of the smartphone group (all *p* < 0.001).

### 3.3. Factors Associated with the Severity of Locomotive Syndrome

The results of the generalized linear model with the GLFS-25 total score and each subdomain score were divided into the three structures as dependent variables and smartphone use, sex, and age as explanatory variables were shown in Table 3. Smartphone use was significantly associated with lower GLFS-25 total and subdomain scores after controlling for sex and age (all *p* < 0.001).

## 4. Discussion

This study explored whether smartphone users are less likely to have “LoS” among Japanese elderly individuals with musculoskeletal disorders. We found that the smartphone group had significantly lower total GLFS-25 scores, body pain, movement-related difficulty, and social and psychosocial complications than the non-smartphone group. This study is the first to explore the relationship between LoS and smartphone use among elderly individuals with musculoskeletal disorders, although the causal relationship remains unclear.

Smartphones are convenient to use, have a variety of functions [16], and provide user satisfaction in several content areas, including socializing, entertainment, and information seeking [17]. The use of smartphones is associated with better cognitive function among elderly individuals [18], and octogenarians who use the Internet have higher cognitive function than those who do not use these phones [19]. Mutual relationships between cognitive function and physical function [20] or LoS [21] may partly explain the results of the present study.

Interestingly, we found a significant difference in the GLSF-25 subdomain score for body pain between the smartphone and non-smartphone groups. However, in this study, a multifaceted and comprehensive assessment of pain was not performed, and the relationship between local pain and smartphone use remains unknown. Further studies are required to confirm this finding.

In this study, we also found a significant difference in movement-related difficulties between the smartphone and non-smartphone groups. The use of smartphone apps has been reported to be effective in improving physical activity and function in patients with chronic low back pain [22]. Because numerous kinds of apps for monitoring an individual’s daily activity levels are freely downloaded onto a smartphone, self-monitoring behavior may be useful for maintaining physical movement components among smartphone users [23,24].

SNS such as Facebook, LINE, and Twitter are popular smartphone apps. These SNS were recently found to be positively associated with life satisfaction, regardless of the number of SNS friends, among all generations in Japan [25]. It is also positively correlated with social connections [6]. Although we did not investigate SNS use among the smartphone group in this study, SNS use may have contributed to their social participation, leading to differences in the GLFS-25 total and psychosocial complication scores from the non-smartphone group.

The current study had some limitations. First, the types of apps, frequency of use, and use of SNS apps in the smartphone group were not examined. It is necessary to clarify whether the types of apps used by smartphone users affect locomotive syndrome and whether the amount of activity differs between smartphone users and non-smartphone users. Second, the number of comorbidities and medications taken by each participant were unknown. Third, the definition of LoS or its severity was assessed only by the GLSF-25 and not by the two other physical function tests (stand-up test and/or two-step test). It is important to note that the prevalence of LoS differs according to the measurement tools [9]. Fourth, this was a cross-sectional study; therefore, the causal relationship between smartphone use and LoS remains unclear. Further longitudinal investigations are necessary to determine whether smartphone use prevents the development of LoS or improves its severity among elderly individuals with musculoskeletal disorders.

## 5. Conclusions

This study tested the hypothesis that the prevalence of LoS among smartphone users would be lower than among non-smartphone users in older adults with musculoskeletal disorders. Results showed that LoS prevalence was significantly higher in the non-smartphone group than in the smartphone group at all LoS stages. Mean GLFS-25 total and subdomain scores were also significantly lower in the smartphone group than in the non-smartphone group, and these statistical differences were maintained even after controlling for age and sex. However, since the causal relationship remains unclear, further longitudinal investigations are required to explore what apps of smartphone users are associated the prevalence or the severity of LoS.

## Figures and Tables

**Figure 1 ijerph-19-16213-f001:**
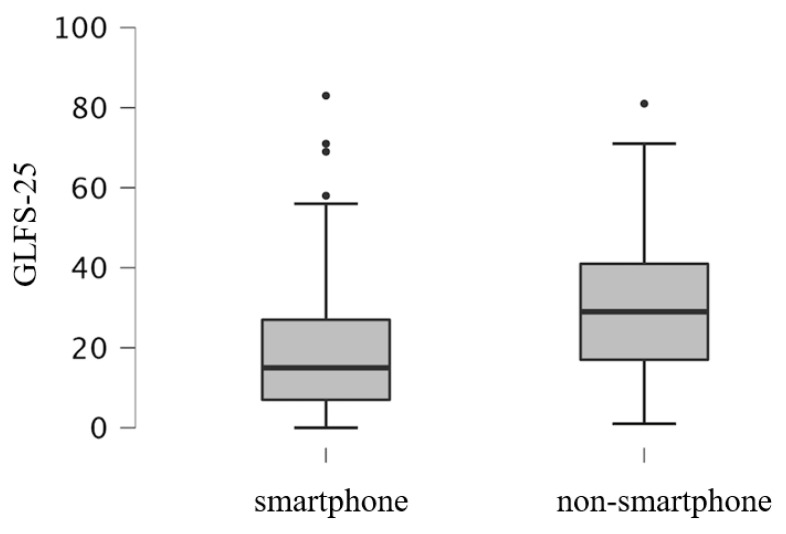
A box plot of the total GLFS-25 scores.

**Figure 2 ijerph-19-16213-f002:**
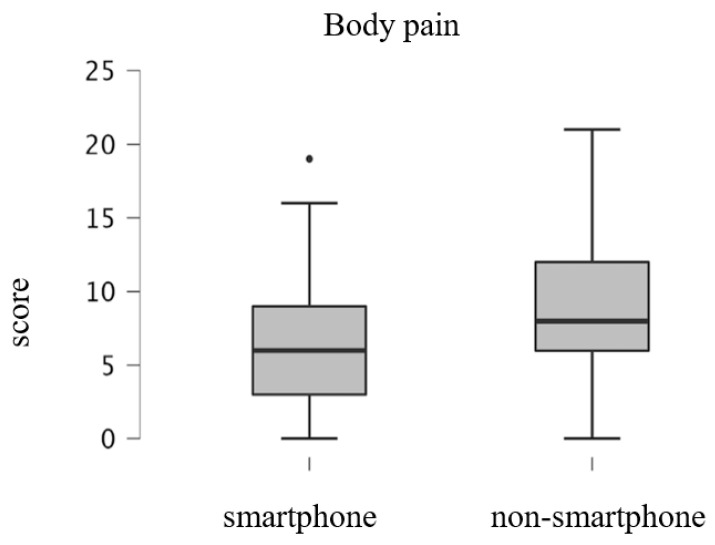
Box plots of the GLFS-25 subdomain score; body pain.

**Figure 3 ijerph-19-16213-f003:**
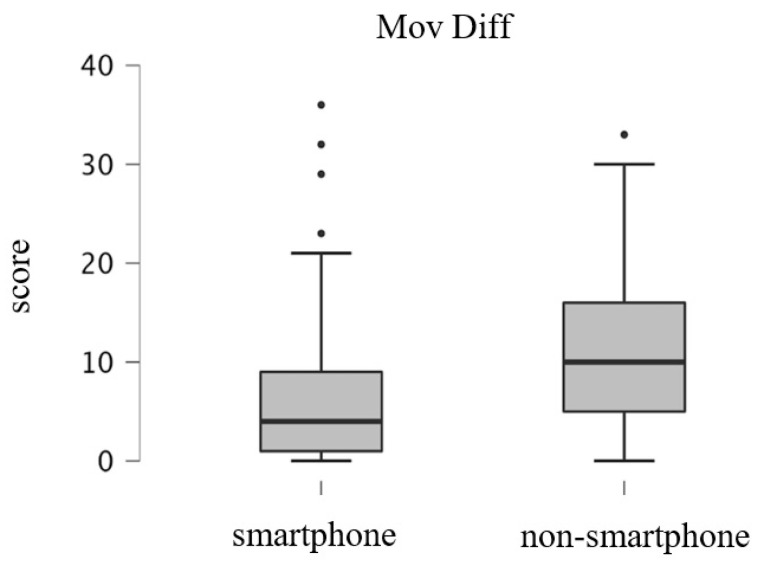
Box plots of the GLFS-25 subdomain score; movement-related difficulty.

**Figure 4 ijerph-19-16213-f004:**
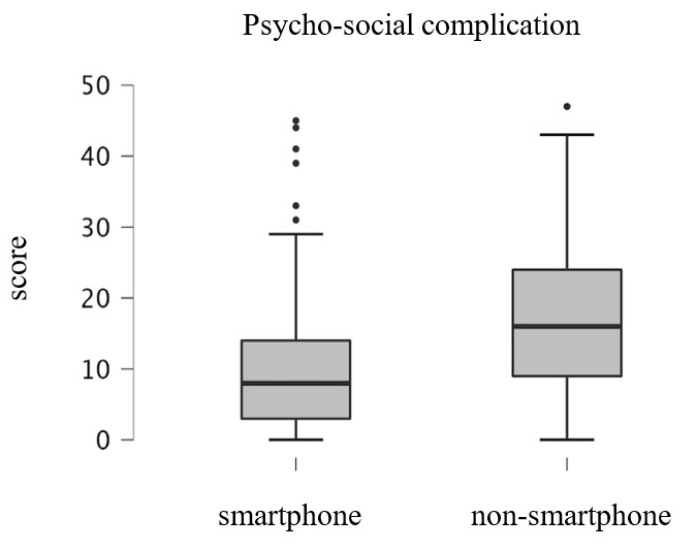
Box plots of the GLFS-25 subdomain score; psycho-social complication.

**Table 1 ijerph-19-16213-t001:** Characteristics of participants.

	Total	Smartphone User (*n* = 133)	Non-Smartphone User (*n* = 133)	*p*-Value
Age (years)	74.7 ± 6.4	72.9 ± 6.3	76.4 ± 6.0	<0.001
male/female	95/171	48/85	47/86	0.898
Reason for orthopedic consultation				
Hip/knee OA	39	20	19	0.484
RA or PMR	82	46	36	
Spinal Disease	124	56	68	
Other	21	11	10	

Hip/knee OA, Hip/Knee osteoarthritis; RA or PMR, rheumatoid arthritis or polymyalgia rheumatica.

**Table 2 ijerph-19-16213-t002:** Prevalence of locomotive syndrome at each stage.

	Smartphone User (*n* = 133)	Non-Smartphone User (*n* = 133)	*p*-Value
LoS stage ≥ 1	105 (78.9%)	125 (94.0%)	<0.001
LoS stage ≥ 2	60 (45.1%)	102 (76.7%)	<0.001
LoS stage 3	42 (31.6%)	80 (60.2%)	<0.001

LoS, locomotive syndrome.

**Table 3 ijerph-19-16213-t003:** Generalized linear models for the GFLS-25 total and subdomain scores.

Parameter	B	Standard Error	95% CI for B		*p*-Value
**Total score**					
Gender (male)	0.015	0.095	−0.173	0.203	0.878
age	−0.001	0.007	−0.016	0.013	0.867
Non-Smartphone User	0.228	0.046	0.137	0.320	<0.001
**Subdomain score**					
*Body pain*					
Gender (male)	0.010	0.080	−0.147	0.166	0.904
age	−0.002	0.006	−0.010	0.015	0.721
Non-Smartphone User	0.302	0.080	0.459	0.144	<0.001
*Movement difficulty*					
Gender (male)	0.206	0.106	−0.003	0.416	0.053
age	−0.004	0.008	−0.021	0.014	0.684
Non-Smartphone User	0.398	0.107	0.609	0.187	<0.001
*Psycho-social complication*					
Gender (male)	0.036	0.099	−0.231	0.721	0.721
age	−0.002	0.008	0.014	0.822	0.822
Non-Smartphone User	0.458	0.100	−0.261	0.000	<0.001

B: Partial regression coefficient; CI: Confidence interval; GFLS-25, 25-question Geriatric Locomotive Function Scale.

## Data Availability

The data used to this study are available from the corresponding author upon request.

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
