# Peer review of "Smartphone Use Is Associated with Low Prevalence of Locomotive Syndrome among Elderly Individuals with Musculoskeletal Disorders"

_ijerph, 2022, doi:10.3390/ijerph192316213_

Round 1

Reviewer 1 Report

Thank you very much for inviting me to review this manuscript. This study aimed at exploring the association between smartphone use and prevalence of locomotive syndrome in seniors with musculoskeletal disorders. My major concerns are as follows:

1. There is lacking information regarding smartphone use affecting health or musculoskeletal disorders in positive or negative manners, thus gaps of the study are unclear.

2. This study recruited participants and asked them to report their locomotive function using the questionnaire but written informed consents were not given from the participants. I think this may aganist the human ethics.

3. I agree with study's limitations that some important details or parameters are not collected, such as types of application during smartphone use, experience and time spending to use a smartphone. Hence, association between smartphone use and level of locomotive syndrome may be interfered by such factors. At the first time when I read abstract, I wished I could found them but no details were presented. It may decrease work's impression and important.

4. The discussion does not explain enough why the associations of locomotive syndrome resulted from musculoskeletal disorders and smartphone use are shown.

5. I do not see Tables in the manuscript.  

Reviewer 2 Report

The study examined the relationship between phone use and musculoskeletal problems in the elderly population.In the introduction part of the study, why this subject needs to be investigated and what gap in the literature.The following studies should be examined and the introduction part needs to be developed a little more.

1-Toh, S. H., Coenen, P., Howie, E. K., & Straker, L. M. (2017). The associations of mobile touch screen device use with musculoskeletal symptoms and exposures: A systematic review. PloS one12(8), e0181220.

2-Zirek, E., Mustafaoglu, R., Yasaci, Z., & Griffiths, M. D. (2020). A systematic review of musculoskeletal complaints, symptoms, and pathologies related to mobile phone usage. Musculoskeletal Science and Practice49, 102196.

3-Eitivipart, A. C., Viriyarojanakul, S., & Redhead, L. (2018). Musculoskeletal disorder and pain associated with smartphone use: A systematic review of biomechanical evidence. Hong Kong Physiotherapy Journal38(02), 77-90.

Method and results section is clear and understandable.

The discussion part should be written by discussing the findings in the light of the literature.The discussion part should be rewritten. The resources used are very limited.

Reviewer 3 Report

It is an interesting paper; I think the use of smartphones because they have limitations to mobile due to body pain or musculoskeletal disorders, but this paper shows that smartphone users have the low locomotive syndrome. The discussion explained that users of smartphones used such exercise applications to maintain their physical activities. But this paper has not asked about the purpose of smartphones among elderly users, so it is necessary to give more explanation about the fact of the respondents and why they have low locomotive syndromes than to cite other research. The conclusion needs suggestions for the next researchers or for the elderly as the respondents about using the smartphone.

Reviewer 4 Report

Thank you for the opportunity to review your manuscript, Smartphone use is associated with low prevalence of locomotive syndrome among elderly individuals with musculoskeletal disorders.

Line 46. Ranks are displayed from youngest to oldest (59-65 years).

Line 56-58. The locomotor syndrome is introduced very lightly—more data or information (prevalence, pathologies included, etc.). 

Line 68-70. The fact that the information on the study is published does not exclude the signing of the informed consent. It is one thing to inform and another to authorise participation in the study and data transfer.

Line 76. Remove the name of the authors who recruited the sample. It is unnecessary to personalise who carried out each part of the study. That information is named in the authors' contribution section. Moreover, data recruitment does not imply the right to conceptual authorship. Therefore, I would remove the word authors and replace it with evaluators or researchers.

Is the GLFS-25 assessment tool valid for classifying subjects, or did the authors create it? It is necessary to use validated tools.

The difference in the age of the groups is significant, and although the difference is only 3.5 years, we cannot rule out that this has an influence. Age is known to affect mobility. There are almost twice as many in the 3

Line 177. I would remove the word demonstrates and replace it with explores, as the data may be conditioned by age. It would NOT be as strong.

Limitations should include the difference in the age of the subjects, which may have influenced the results.

"Figure 2" should be split into three separate figures.

Round 2

Reviewer 1 Report

No revision needed.

Author Response

I have spell-checked the text and will submit the text again.

Reviewer 2 Report

No comments.

Author Response

(The authors gave the same response as above.)

Reviewer 4 Report

The authors have responded to all my clarifications.

Author Response

(The authors gave the same response as above.)
